# Time-Dependent Effective Hamiltonians for Light–Matter Interactions

**DOI:** 10.3390/e26060527

**Published:** 2024-06-19

**Authors:** Aroaldo S. Santos, Pedro H. Pereira, Patrícia P. Abrantes, Carlos Farina, Paulo A. Maia Neto, Reinaldo de Melo e Souza

**Affiliations:** 1Instituto de Física, Universidade Federal Fluminense, Niterói 24210-346, Rio de Janeiro, Brazil; arofisica@gmail.com (A.S.S.); pedro_h@id.uff.br (P.H.P.); reinaldos@id.uff.br (R.d.M.e.S.); 2Instituto Federal do Paraná, Telêmaco Borba 84269-090, Paraná, Brazil; 3Instituto de Física, Universidade Federal do Rio de Janeiro, Rio de Janeiro 21941-972, Rio de Janeiro, Brazil; ppabrantes91@gmail.com (P.P.A.); farina@if.ufrj.br (C.F.)

**Keywords:** effective Hamiltonians, time-dependent Hamiltonians, quantum fluctuations, molecular quantum electrodynamics, light–matter interactions

## Abstract

In this paper, we present a systematic approach to building useful time-dependent effective Hamiltonians in molecular quantum electrodynamics. The method is based on considering part of the system as an open quantum system and choosing a convenient unitary transformation based on the evolution operator. We illustrate our formalism by obtaining four Hamiltonians, each suitable to a different class of applications. We show that we may treat several effects of molecular quantum electrodynamics with a direct first-order perturbation theory. In addition, our effective Hamiltonians shed light on interesting physical aspects that are not explicit when employing more standard approaches. As applications, we discuss three examples: two-photon spontaneous emission, resonance energy transfer, and dispersion interactions.

## 1. Introduction

In molecular quantum electrodynamics, atoms and molecules are treated within non-relativistic quantum mechanics, and the electromagnetic field mediating the interactions is quantized. This approach, born with Dirac’s seminal treatment of spontaneous emission [1], is still an ongoing and intense research field, especially with the unprecedented control of light–matter interactions at the atomic scale reached in the last decades.

All phenomena in this topic can be fully understood by starting with the classical minimal coupling Hamiltonian and quantizing it. For molecular quantum electrodynamics, the most convenient approach is to work in the Coulomb gauge. Throughout this work, we shall deal with neutral molecules, in which case we can make a unitary transformation on the minimal coupling Hamiltonian and work with the equivalent multipolar Hamiltonian [2,3,4,5,6]. Nonetheless, the generality of these Hamiltonians is also their main weakness, since we must perform extensive calculations to obtain the quantities describing most of the effects. For instance, the interaction between two nonpolar molecules in their ground state results from a tedious fourth-order perturbative calculation.

Here comes the convenience of working with effective Hamiltonians, which are tailored for each specific application, bringing several physical insights and shortening the technical calculation to a much simpler and lower perturbative order analysis. An insightful example is the dynamical polarizability (DP) Hamiltonian, obtained by R. Passante and collaborators [7,8], which is built directly on the molecular dynamical polarizability instead of its electric dipole operator, capturing better the physics governing the interaction. Indeed, nonpolar molecules do not possess permanent electric dipole moments, and their interaction is possible only due to virtual internal transitions that are automatically taken into account by the dynamical polarizability. This is the main message of effective Hamiltonians: building the relevant physical mechanism into the Hamiltonian lowers the perturbative order required in calculations. With the DP Hamiltonian, intermolecular interactions are determined by means of a second-order calculation. Effective Hamiltonians and actions are also useful in describing non-stationary systems and have been employed to develop a multipolar approach to the dynamical Casimir effect [9] and to understand its microscopic origin [10,11,12,13,14].

Effective Hamiltonians are easily constructed from unitary transformations [15], but there is no general recipe to generate useful ones. In this paper, we fill this gap by presenting a systematic method to obtain convenient effective Hamiltonians and by discussing their physical implications. To illustrate our approach, we derive four general effective Hamiltonians allowing us to extend the scope of applications to important phenomena in molecular quantum electrodynamics.

Our method is based on choosing a unitary transformation inspired by (but not equal to) the Hermitian conjugate of the evolution operator for the system. A key concept for our formalism is that the linear susceptibility χ of a quantum system is built from the unequal time commutator of an appropriate operator O describing the system. For the context explored in this paper, we shall take O as being (i) the molecular dipole operator, in which case χ is related to the molecular polarizability, and (ii) the electric field operator, with χ representing the electric field generated by a point dipole (Green function), as discussed in Appendix A.

The importance of the unequal time commutators of the electric field operators in connection to the measurability of the fields was stressed in the literature [16,17]. Here, we show that the unequal time commutators can be taken as the basis to generate convenient effective Hamiltonians by allowing a given degree of freedom to effectively dress another one. Physically, this is equivalent to considering part of our system (a molecule or the electromagnetic field) as an open quantum system that is effectively dressed by an appropriate unitary evolution operator, thus yielding an effective time-dependent Hamiltonian for the system.

In Section 2, we employ our formalism to set up the Hamiltonian $H_{M}^{\mathrm{eff}}$, in which the molecular degrees of freedom are dressed by the field. This Hamiltonian generalizes the DP one in two aspects: (i) it naturally accounts for internal dissipation in the molecules, and (ii) it does not require the molecules to remain in the same internal state (usually the ground state) during the process to be described. The latter aspect is a key element and enables us to evaluate the two-photon spontaneous emission (TPSE) in Section 3 within first-order perturbation theory—a much simpler route than the one commonly followed in the literature.

From Section 4 on, we work with situations involving more than one molecule. We employ our method to build the second effective Hamiltonian $H_{F}^{\mathrm{eff}}$, where one molecule, say molecule *B*, dresses the field. With this dressing, the field acting on the other molecules is given by the superposition of the vacuum electric field with the electric dipole field generated by *B*. In Section 5, we demonstrate the convenience of $H_{F}^{\mathrm{eff}}$ by directly computing the resonance energy transfer (RET) between two quantum emitters in first order.

Then, in Section 6, we analyze dipole–dipole correlation effects and show that different effective Hamiltonians are convenient depending on the distance separating the molecules. In the asymptotic long-distance limit, we demonstrate the Hamiltonian $H_{\mathrm{MF}}^{\mathrm{eff}}$, in which the field dresses the molecules, and, in turn, one of the dressed molecules dresses back the field. This Hamiltonian is similar to $H_{F}^{\mathrm{eff}}$, but now the electric field generated by molecule *B* does not depend on the molecular dipole operator. Instead, it is produced by the dipole induced by the vacuum itself. In the particular case where we assume the molecules to be in the ground state, in the long-distance limit, we recover the Hamiltonian originally proposed by P.W. Milonni [18].

Finally, for the short-distance limit (the non-retarded regime), we follow a complementary route: first, one molecule dresses the field, and then the dressed field dresses back the molecules. This leads us to a new effective Hamiltonian $H_{\mathrm{FM}}^{\mathrm{eff}}$. We demonstrate its convenience by applying it in Section 7 to obtain the London interaction energy in first-order perturbation theory and show that this route provides some relevant physical insights. Indeed, our approach can quantify the contributions to the interaction energy coming from the dipole fluctuations of each molecule. The above examples surely do not exhaust the list of useful effective Hamiltonians, and our method should be valuable for several additional applications.

## 2. The Field Dresses the Molecules

We begin with a single neutral molecule at position R in the presence of the quantized electromagnetic field. In the dipole approximation, the Hamiltonian describing the system is
(1)$$H_{\mathrm{total}}=H_{0}-d\cdot E\left(R\right)\;,$$
where d is the molecular dipole operator, and E(R) is the quantized electric field evaluated at the molecule’s center-of-mass position R. Note that in the dipole approximation, the electric field can be taken as uniform over the scale of the molecule. $H_{0}$ stands for the free Hamiltonians and is given by
(2)$$H_{0}=H_{0m}+\sum_{k\sigma }\hbar \omega_{k}\left(a_{k\sigma }^{†}a_{k\sigma }+\frac{1}{2}\right)\;.$$
In this expression $a_{k\sigma }\left(a_{k\sigma }^{†}\right)$ stands for the annihilation (creation) operator for a photon with wavector k and polarization σ, whose electromagnetic field oscillates with a frequency $\omega_{k}=\left|k\right|c$. $H_{0m}$ is the free molecular Hamiltonian whose eigenstates and eigenenergies are assumed to be known. The coupling between the molecule and field, given by the second term on the right-hand side of Equation (Equation 1), can be treated as a perturbation. Therefore, it is convenient to work in the interaction picture with the interaction Hamiltonian
(3)H(t)=−d(t)·E(R,t).
The time dependence is obtained by evolving the operators with the free Hamiltonian $H_{0}$. A nonpolar molecule is characterized by not having a permanent electric dipole in its ground state |g〉, i.e., 〈g|d|g〉=0. Thus, any process during which the molecule does not excite, such as the Stark shift, must be obtained at least through second-order perturbation theory. If |ψ(t)〉 symbolizes the state of the molecule–field system in the interaction picture, then its evolution can be written as
(4)$$i\hbar \frac{d}{dt}|\psi \left(t\right)\rangle =H\left(t\right)|\psi \left(t\right)\rangle \;.$$

An equivalent description is generated once we apply a unitary transformation to the state. We choose it as
(5)$$U_{M}\left(t\right)=e^{\frac{i}{\hbar }\int_{-\infty }^{t}\;\;\;\;dt^{\prime }H\left(t^{\prime }\right)}\;.$$
Note that transformation (Equation 5) implements the Heisenberg picture to first order in *H*, thus canceling, at this order, the time evolution of |ψ(t)〉, which is precisely our goal. If $[H\left(t\right),H\left(t^{\prime }\right)]=0$, then the transformation given in Equation (Equation 5) would implement the Heisenberg picture exactly. Therefore, in the representation defined by $U_{M}$, the time evolution of |ψ(t)〉 results from the non-vanishing value of the commutator $\left[H\left(t\right),H\left(t^{\prime }\right)\right]$, which is consistent with the discussion on linear susceptibilities outlined in Section 1. As will become clear below, this transformation effectively dresses the molecular degree of freedom indicated by the subscript M. Next, we derive the equation satisfied by $|\psi_{M}\left(t\right)\rangle =U_{M}|\psi \left(t\right)\rangle$. From Equation (Equation 4), we find
(6)$$i\hbar \frac{d}{dt}|\psi_{M}\left(t\right)\rangle =H_{M}|\psi_{M}\left(t\right)\rangle \;,$$
with
(7)$$H_{M}\left(t\right)=U_{M}\left(t\right)H\left(t\right)U_{M}{\left(t\right)}^{-1}+i\hbar \frac{dU_{M}\left(t\right)}{dt}U_{M}^{-1}\left(t\right)\;.$$
Here enters the fact that we are not looking for an equivalent Hamiltonian but rather an effective one. We desire an equivalent Hamiltonian only up to quadratic order in the dipole operator, and thus, we are allowed to expand $U_{M}$ and collect results up to the second order in *H*, obtaining
(8)$$\begin{array}{c} U_{M}\left(t\right)\approx 1+\frac{i}{\hbar }\int_{-\infty }^{t}\;\;\;\;dt^{\prime }H\left(t^{\prime }\right)-\frac{1}{2\hbar^{2}}{\left[\int_{-\infty }^{t}\;\;\;\;dt^{\prime }H\left(t^{\prime }\right)\right]}^{2}, \end{array}$$
(9)$$\begin{array}{c} \frac{d}{dt}U_{M}\left(t\right)\approx \frac{i}{\hbar }H\left(t\right)-\frac{1}{2\hbar^{2}}\int_{-\infty }^{t}\;\;\;\;dt^{\prime }\left\{H\left(t\right),H\left(t^{\prime }\right)\right\}\;. \end{array}$$
Note that the expansion in Equation (Equation 8) does not correspond to a Dyson series, since the unitary transformation given in Equation (Equation 5) is not an evolution operator (it lacks a time-ordering operator). These expansions differ in the second-order term, and we stress that the third term on the right side of Equation (Equation 8) is proportional to the square of the second term, which is crucial for the results we will obtain. We define the effective Hamiltonian $H_{M}^{\mathrm{eff}}\left(t\right)$ as the second-order approximation of $H_{M}$, which is obtained by substituting the previous relations into Equation (Equation 7):(10)$$H_{M}^{\mathrm{eff}}\left(t\right)=-\frac{i}{2\hbar }\int_{-\infty }^{t}\;\;\;\;dt^{\prime }\left[H\left(t\right),H\left(t^{\prime }\right)\right]\;,$$
where we used the identity $2H\left(t\right)H\left(t^{\prime }\right)=\left[H\left(t\right),H\left(t^{\prime }\right)\right]+\left\{H\left(t\right),H\left(t^{\prime }\right)\right\}$. Notice that the linear term in the dipole vanished. From Equation (Equation 3) and since electric field operators at the same spatial point commute at all times (see Appendix A), we are left with
(11)$$H_{M}^{\mathrm{eff}}\left(t\right)=-\frac{i}{2\hbar }\int_{-\infty }^{t}\;\;\;\;dt^{\prime }\left[d_{j}\left(t\right),d_{l}\left(t^{\prime }\right)\right]E_{j}\left(R,t\right)E_{l}\left(R,t^{\prime }\right)\;,$$
where we employed Einstein notation and denoted by j,l=1,2,3 the Cartesian components of the operators. The great convenience of this Hamiltonian is that it is quadratic in the operators, thus halving the required perturbation order in comparison to the Hamiltonian given by Equation (Equation 3). We point out that our demonstration remains the same whether the electric field is quantized or not. As an example, with this effective Hamiltonian, the Stark effect can be obtained from a first-order perturbative calculation. We emphasize that this Hamiltonian is valid only within first-order perturbation theory, but improvements can be made if one keeps extra terms in Equations (Equation 8) and (9). $H_{M}^{\mathrm{eff}}$ mixes both the materials’ and fields’ degrees of freedom. When the atom is assumed to remain in the ground state, we may take the expectation value of $H_{M}^{\mathrm{eff}}$ in the molecular’s subspace defined by the ground state through the evaluation of $\langle g|H_{M}^{\mathrm{eff}}\left(t\right)|g\rangle$. We stress here that we are not acting on the field subspace, and thus, this average is still an operator in the field variables, which we denote by
(12)$$H_{M}^{\mathrm{eff}(gg)}\left(t\right)=-\frac{1}{2}\int_{-\infty }^{\infty }\;\;\;\;dt^{\prime }\alpha_{jl}\left(t-t^{\prime }\right)E_{j}\left(R,t\right)E_{l}\left(R,t^{\prime }\right)\;,$$
with
(13)$$\alpha_{jl}\left(t-t^{\prime }\right)=\frac{i}{\hbar }\theta \left(t-t^{\prime }\right)\langle g|[d_{j}\left(t\right),d_{l}\left(t^{\prime }\right)]|g\rangle$$
being the molecular dynamical polarizability tensor for the ground state describing its linear response to an applied electric field—see Appendix A for details. For practical applications and some physical interpretations, it is generally more suitable to work with the dynamical polarizability in the Fourier representation rather than in the time domain. To do so, we write the free electromagnetic field in the usual form:(14)$$\begin{array}{cc} E(R,t^{\prime }) & =\sum_{k\sigma }E_{k\sigma }\left(R,t^{\prime }\right)=\sum_{k\sigma }\left[E_{k\sigma }^{\left(+\right)}\left(R\right)e^{-i\omega_{k}t^{\prime }}+E_{k\sigma }^{\left(-\right)}\left(R\right)e^{i\omega_{k}t^{\prime }}\right], \end{array}$$
where $\omega_{k}=c\left|k\right|$, σ is the polarization degree of freedom, and the superscript + (−) refers to positive (negative) frequencies of the field. Substituting Equation (Equation 14) into (Equation 12), we arrive at
(15)$$H_{M}^{\mathrm{eff}(gg)}\left(t\right)=-\frac{1}{2}d^{\mathrm{ind}}\left(t\right)\cdot E\left(R,t\right)\;,$$
where
(16)$$d_{j}^{\mathrm{ind}}\left(t\right)=\sum_{k\sigma }\left[\alpha_{jl}\left(\omega_{k}\right)E_{k\sigma }^{l(+)}\left(R,t\right)+\alpha_{jl}\left(-\omega_{k}\right)E_{k\sigma }^{l(-)}\left(R,t\right)\right]$$
stands for the vacuum-induced dipole operator, and $E_{k\sigma }^{l(\pm )}\left(R,t\right)=E_{k\sigma }^{l(\pm )}\left(R\right)e^{\mp i\omega_{k}t}$ is the *l*-th Cartesian component of $E_{k\sigma }^{\left(\pm \right)}\left(R,t\right)$. Notice that $d^{\mathrm{ind}}\left(t\right)$ acts on the field’s Hilbert space. Due to the reality of $\alpha (t-t^{\prime })$, $\alpha_{jl}\left(-\omega_{k}\right)=\alpha_{jl}^{*}\left(\omega_{k}\right)$, and, thus, $d^{\mathrm{ind}}\left(t\right)$ is an Hermitian operator. If dissipation is negligible, we re-obtain as a particular case the DP Hamiltonian [7]
(17)$$H_{M}^{\mathrm{eff}(\mathrm{DP})}\left(t\right)=-\frac{1}{2}\sum_{k\sigma }\alpha_{jl}\left(\omega_{k}\right)E_{k\sigma }^{l}\left(R,t\right)E^{j}\left(R,t\right)\;.$$
Physically, this is the quantum counterpart of the interaction energy of a polarizable system without permanent electric dipole moments in the presence of an external electric field. In the case without dissipation, the dynamical polarizability in the Fourier space is given by (see Appendix A)
(18)$$\alpha_{jl}\left(\omega \right)=-\frac{1}{\hbar }\sum_{r\neq g}d_{j}^{gr}d_{l}^{rg}\left(\frac{1}{\omega -\omega_{rg}}-\frac{1}{\omega +\omega_{rg}}\right)\;,$$
where *r* denotes the excited internal molecular states, and $d^{gr}=\langle g|d|r\rangle =d^{rg*}$ is the transition dipole moment between states *g* and *r*, while $\omega_{rg}$ is the corresponding transition frequency.

There are some subtleties concerning the unitary transformation employed in this section. One could argue that once the integration present in Equation (Equation 5) starts from −∞, our truncation in Equation (9) is not rigorous. The point is that the molecule has a finite memory, characterized by a time scale τ. This means that $\alpha_{jl}\left(t-t^{\prime }\right)$ vanishes for $t-t^{\prime }\gg \tau$ in Equation (Equation 13), enabling the lower limit in (Equation 5) to be replaced by t−τ. The validity of this truncation is then tantamount to the validity of the perturbative method in the molecule–field interaction, justifying our approach. This argument also underlies the convenience of working in the Fourier space. Convergence of the time integration in Equation (Equation 12) requires that we account for dissipation in the polarizability. Nevertheless, in many cases of interest, the most relevant Fourier modes are far from molecular resonances, and we may neglect dissipation when using Equations (Equation 15) and (Equation 16).

Another important aspect is that the effective Hamiltonian (Equation 11) is convenient only when first-order perturbation theory in Hamiltonian (Equation 3) vanishes, even though regularization techniques may render it applicable if this is not the case [8]. We may separate the main applications of the effective Hamiltonian $H_{M}^{\mathrm{eff}}$ into two groups: (i) the molecule remains in the same internal state during the entire process, and the expectation value of the electric dipole operator in this state is zero; (ii) the molecule undergoes a transition between two internal states, but the electric dipole operator is unable to connect these two states. Examples involving (i) have already been discussed in the literature [7,8], in contrast with case (ii). One fascinating example of this second group is the two-photon spontaneous emission, with selection rules forbidding the one-photon transition. In the next section, we explore this example from the perspective of the effective Hamiltonian derived in this section.

## 3. Application to the Two-Photon Spontaneous Emission

An excited molecule may decay to its ground state through the emission of two photons in the so-called two-photon spontaneous emission (TPSE). This phenomenon is particularly interesting when the one-photon transition is forbidden. The TPSE makes the vacuum unstable and is responsible for the initial buildup of the intracavity field in two-photon micromasers [19,20,21]. More recently, it was shown that the simultaneously emitted photons can be indistinguishable and entangled in time and frequency [22,23,24], renewing the interest in this phenomenon [25,26,27,28,29]. This section aims to obtain the TPSE rate directly from first-order perturbation theory in Hamiltonian (Equation 11). Let us consider that a molecule in an internal state |e〉 decays in vacuum to its ground state |g〉 through the emission of two photons with wavevectors k and $k^{\prime }$ and polarizations σ and $\sigma^{\prime }$. To this end, it suffices to analyze the matrix element of $H_{M}^{\mathrm{eff}}$ connecting the initial and the final states, given by
(19)$$\langle g;1_{k\sigma }1_{k^{\prime }\sigma^{\prime }}|H_{M}^{\mathrm{eff}}\left(t\right)|e;0\rangle =-\frac{1}{2}\int_{-\infty }^{\infty }\;\;\;\;dt^{\prime }D_{jl}^{ge}\left(t,t^{\prime }\right)\langle 1_{k\sigma }1_{k^{\prime }\sigma^{\prime }}|E_{j}\left(0,t\right)E_{l}\left(0,t^{\prime }\right)|0\rangle \;,$$
where we chose the origin of our coordinate system at the position of the molecule. Here, |0〉 denotes the vacuum state of the electromagnetic field. We also define
(20)$$D_{jl}^{ge}\left(t,t^{\prime }\right)=\frac{i}{\hbar }\theta \left(t-t^{\prime }\right)\langle g|\left[d_{j}\left(t\right),d_{l}\left(t^{\prime }\right)\right]|e\rangle \;,$$
which involves only the molecular degrees of freedom and quantifies the linear response of the molecule to an applied field connecting internal states |e〉 and |g〉. Note that, when taking |e〉=|g〉 in Equation (Equation 20), the tensor $\overset{↔}{D}$ yields as a particular case the polarizability of the molecule, which is given by Equation (Equation 13). The TPSE rate is immediately obtained in the long-time limit by substituting Equation (Equation 19) into Fermi’s golden rule. In general, it is more convenient to represent $\overset{↔}{D}$ in Fourier space. We begin by writing
(21)$$d\left(t\right)=e^{iH_{0m}\left(t-t_{0}\right)}de^{-iH_{0m}\left(t-t_{0}\right)}$$
and, at the end, we take $t_{0}\to -\infty$. In this expression, $H_{0m}$ denotes the free molecular Hamiltonian, with eigenstates satisfying $H_{0m}|r\rangle =\hbar \omega_{r}|r\rangle$, so that by inserting a closure relation $I=\sum_{r}\left|r\rangle \langle r\right|$ into Equation (Equation 20), we obtain
(22)$$D_{jl}^{ge}\left(t,t^{\prime }\right)=\alpha_{jl}^{ge}\left(t-t^{\prime }\right)e^{-i\omega_{eg}t^{\prime }}\;,$$
with
(23)$$\begin{array}{cc} \alpha_{jl}^{ge}\left(t-t^{\prime }\right) & =\frac{i}{\hbar }\theta \left(t-t^{\prime }\right)\sum_{r}\left[d_{j}^{gr}d_{l}^{re}e^{-i\omega_{rg}\left(t-t^{\prime }\right)}-d_{l}^{gr}d_{j}^{re}e^{i\omega_{re}\left(t-t^{\prime }\right)}\right]. \end{array}$$
The instant $t_{0}$ plays a role only in an unimportant global phase, which was discarded. In Fourier space, Equation (Equation 23) becomes
(24)$$\alpha_{jl}^{ge}\left(\omega \right)=\frac{1}{\hbar }\sum_{r}\left(\frac{d_{j}^{gr}d_{l}^{re}}{\omega_{rg}-\omega }+\frac{d_{l}^{gr}d_{j}^{re}}{\omega_{re}+\omega }\right)\;.$$
The desired matrix element given in Equation (Equation 19) can be obtained from Equation (Equation 14). For emission processes, only positive frequency modes contribute. Using Equation (Equation 22), we also obtain
(25)$$\begin{array}{ccc} \langle g;1_{k\sigma }1_{k^{\prime }\sigma^{\prime }}|H_{M}^{\mathrm{eff}}\left(t\right)|e;0\rangle & & =\frac{\hbar \sqrt{\omega_{k}\omega_{k^{\prime }}}}{4\varepsilon_{0}V}e^{i(\omega_{k}+\omega_{k^{\prime }}-\omega_{eg})t}\times \\ & & \left[\epsilon_{k\sigma }^{j}\epsilon_{k^{\prime }\sigma^{\prime }}^{l}\alpha_{jl}^{ge}\left(\omega_{eg}-\omega_{k^{\prime }}\right)+\epsilon_{k^{\prime }\sigma^{\prime }}^{j}\epsilon_{k\sigma }^{l}\alpha_{jl}^{ge}\left(\omega_{eg}-\omega_{k}\right)\right]\;, \end{array}$$
where $\epsilon_{k\sigma }^{j}$ is the *j*th Cartesian component of the polarization unit vector for the mode with wavevector k and polarization σ, and *V* is the volume of the quantization box. In the long-time limit, we are interested in photon pairs satisfying the condition $\omega_{k}+\omega_{k^{\prime }}=\omega_{eg}$. With this, we see that $\alpha_{jl}\left(\omega_{eg}-\omega_{k^{\prime }}\right)=\alpha_{jl}\left(\omega_{k}\right)=\alpha_{lj}\left(\omega_{eg}-\omega_{k}\right)$, where we used Equation (Equation 24) in the last equality. Therefore, we may simplify Equation (Equation 25) to
(26)$$\langle g;1_{k\sigma }1_{k^{\prime }\sigma^{\prime }}|H_{M}^{\mathrm{eff}}\left(t\right)|e;0\rangle =\frac{\hbar \sqrt{\omega_{k}\omega_{k^{\prime }}}}{2\varepsilon_{0}V}e^{i(\omega_{k}+\omega_{k^{\prime }}-\omega_{eg})t}\epsilon_{k^{\prime }\sigma^{\prime }}^{j}\epsilon_{k\sigma }^{l}\alpha_{jl}^{ge}\left(\omega_{eg}-\omega_{k}\right)$$The probability rate of emitting one photon in a solid angle dΩ around $\hat{k}$ and with a frequency in the interval (ω,ω+dω) and another in a solid angle $d\Omega^{\prime }$ around ${\hat{k}}^{\prime }$, as well as with a frequency in the interval $\left(\omega^{\prime },\omega^{\prime }+d\omega^{\prime }\right)$, is given by (from now on we denote $\omega \equiv \omega_{k}$ and $\omega^{\prime }\equiv \omega_{k^{\prime }}$)
(27)$$d\Gamma^{\mathrm{TPSE}}=\frac{V^{2}}{{\left(2\pi \right)}^{6}}d\Omega d\Omega^{\prime }d\omega d\omega^{\prime }\frac{\omega^{2}\omega^{\prime 2}}{c^{6}}\frac{{\left|\int_{0}^{t}dt^{\prime }\langle g;1_{k\sigma }1_{k^{\prime }\sigma^{\prime }}|H_{M}^{\mathrm{eff}}\left(t^{\prime }\right)|e;0\rangle \right|}^{2}}{\hbar^{2}t}\;.$$
Employing Fermi’s golden rule, we arrive at
(28)$$\frac{d\Gamma^{\mathrm{TPSE}}}{d\Omega d\Omega^{\prime }d\omega d\omega^{\prime }}=\frac{\omega^{3}\omega^{\prime 3}}{c^{6}{\left(2\pi \right)}^{5}{\left(2\varepsilon_{0}\right)}^{2}}{\left|\epsilon_{k^{\prime }\sigma^{\prime }}^{j}\epsilon_{k\sigma }^{l}\alpha_{jl}^{ge}\left(\omega_{eg}-\omega \right)\right|}^{2}\delta \left(\omega_{eg}-\omega -\omega^{\prime }\right)\;.$$
Integration over the solid angles may be readily evaluated from the identity
(29)$$\sum_{\sigma ,\sigma^{\prime }}\int d\Omega d\Omega^{\prime }\epsilon_{k^{\prime }\sigma^{\prime }}^{j}\epsilon_{k\sigma }^{m*}\epsilon_{k^{\prime }\sigma^{\prime }}^{n*}\epsilon_{k\sigma }^{l}=\frac{{\left(8\pi \right)}^{2}}{9}\delta_{ml}\delta_{jn}\;.$$
We also integrate over $\omega^{\prime }$ to find the photon emission rate:
(30)$$\frac{d\Gamma^{\mathrm{TPSE}}}{d\omega }=\frac{\omega^{3}{\left(\omega_{eg}-\omega \right)}^{3}}{18c^{6}\pi^{3}\varepsilon_{0}^{2}}\alpha_{jl}^{ge}\left(\omega_{eg}-\omega \right)\alpha_{jl}^{*ge}\left(\omega_{eg}-\omega \right)\;,$$
which is equivalent to the result of Ref. [2].

When performing second-order perturbation theory, the usual notation is to describe the molecular response in terms not of $\alpha_{ge}$—which is a function of a single frequency variable, but rather a function of two frequency variables, which are obtained from Equation (Equation 24) by replacing $\omega_{re}+\omega$ by $\omega_{rg}-\omega^{\prime }$ in the second term. The calculation from the new effective Hamiltonian (Equation 11) not only yields the two-photon spontaneous emission rate with a much shorter first-order calculation but also describes the results in terms of a single frequency variable function that sheds an interesting light on the physical mechanism involved in the phenomenon. In order to unveil the physical significance of $\alpha_{jl}^{\mathrm{ge}}$, let us project the Hamiltonian $H_{M}^{\mathrm{eff}}$ into the field’s Hilbert space, thus generalizing Equation (Equation 12) for situations where the molecule undergoes an internal transition. This is done by defining the new effective Hamiltonian from Equation (Equation 11):(31)$$H_{M}^{\mathrm{eff}(ge)}\left(t\right):=\langle g|H_{M}^{\mathrm{eff}}\left(t\right)|e\rangle =-\frac{1}{2}\int_{-\infty }^{t}\;\;\;\;dt^{\prime }D_{jl}^{ge}\left(t,t^{\prime }\right)E_{j}\left(0,t\right)E_{l}\left(0,t^{\prime }\right)\;,$$
which involves only electric field operators. Note that $D_{jl}^{ge}\left(t,t^{\prime }\right)$, given by Equation (Equation 20), is not a real number, and, therefore, $H_{M}^{\mathrm{eff}(ge)}$ is non-Hermitian. This is due to the fact that the field degrees of freedom alone constitute an open quantum system, extracting energy from the drive provided by the molecular internal transitions encapsulated in $D_{jl}^{ge}\left(t,t^{\prime }\right)$. The non-hermiticity of Hamiltonian (31) also reflects the break of time inversion symmetry imposed by the two-photon decay.

Following the same steps that led us from Equations (Equation 12) to (Equation 15), we obtain
(32)$$H_{M}^{\mathrm{eff}(ge)}\left(t\right)=-\frac{1}{2}d^{\mathrm{ind}(ge)}\left(t\right)\cdot E\left(R,t\right)\;,$$
where the induced dipole for the transition |e〉⟶|g〉 is given by
(33)$$\begin{array}{c} d_{j}^{\mathrm{ind}(ge)}\left(t\right)=\sum_{k\sigma }\left[\alpha_{jl}^{ge}\left(\omega_{eg}+\omega_{k}\right)E_{k\sigma }^{l(+)}\left(R,t\right)+\alpha_{jl}^{ge}\left(\omega_{eg}-\omega_{k}\right)E_{k\sigma }^{l(-)}\left(R,t\right)\right]e^{-i\omega_{eg}t}\;. \end{array}$$
For this reason, we denominate $\alpha_{jl}^{ge}$ as the *transition polarizability tensor*. This is a useful concept whenever the transition dipole element of a given internal molecular transition vanishes but can be induced by an external electric field. It generalizes the concept of the polarizability tensor, which stands for the dipole induced for a fixed internal molecular state. This induced transition dipole acts as an external source oscillating with frequency $\omega_{eg}$ and driving the field appearing in the effective Hamiltonian (31). Here, $\omega_{eg}>0$ indicates that energy-conserving processes must be accompanied by photon creation, as can be verified in Equation (Equation 14). In this case, only the last term in Equation (33) contributes to the process. The other term is relevant for two-photon absorption, and the calculation presented in the section applies with minor modifications to this case.

## 4. The Molecules Dress the Field

In the previous section, we investigated the convenience of employing effective Hamiltonians in which the electric field dresses the molecules. Now, we shall analyze the opposite case and present an effective Hamiltonian that describes a molecule dressing the electric field operator. Consider two nonpolar molecules *A* and *B*. The electric dipole Hamiltonian describing this system in the interaction picture is
(34)$$H^{\left(2\right)}=H_{A}+H_{B}\;,$$
where $H_{\zeta }=-d_{\zeta }\left(t\right)\cdot E\left(R_{\zeta },t\right)$, and $d_{\zeta }$ is the electric dipole operator of molecule ζ=A,B, whose center of mass is at position $R_{\zeta }$. We again represent the system’s state with |ψ(t)〉, satisfying Equation (Equation 4), but implicitly including a tensor product of both the molecules’ and fields’ states. We follow the same reasoning as in the previous section. In this case, however, we want the molecule *B* to dress the electric field operator. Hence, we choose as the unitary transformation the inverse of the evolution operator for the coupling between molecule *B* and the field:(35)$$U_{F}=\tilde{T}e^{\frac{i}{\hbar }\int_{-\infty }^{t}\;\;\;\;dt^{\prime }H_{B}\left(t^{\prime }\right)}\;,$$
where $\tilde{T}$ is the anti-time ordering operator (earlier-time operators on the left). Its presence implies a crucial difference in comparison with Equation (Equation 5), and its purpose is to eliminate $H_{B}$ so that the entire role played by molecule *B* will be through the field it produces. If only molecule *B* were present, the unitary transformation $U_{F}$ would take the interaction picture into the Heisenberg picture. Nonetheless, in the presence of atom *A*, this unitary transformation yields a new effective Hamiltonian, to which we now turn our attention.

Following steps analogous to those in Section 2, the equivalent Hamiltonian is given by
(36)$$H_{F}=U_{F}H^{\left(2\right)}\left(t\right)U_{F}^{-1}+i\hbar \frac{\partial U_{F}}{\partial t}U_{F}^{-1}\;.$$
Due to the anti-time ordering operator, we have $i\hbar \partial_{t}U_{F}=-U_{F}H_{B}$, and, thus,
(37)$$H_{F}=-d_{A}\left(t\right)\cdot U_{F}E\left(R_{A},t\right)U_{F}^{-1}\;,$$
canceling $H_{B}$ as anticipated. Expanding up to the linear term in $d_{B}$, we obtain (see Appendix A)
(38)$$U_{F}E\left(R_{A},t\right)U_{F}^{-1}\approx E\left(R_{A},t\right)+E_{\mathrm{dip},B}\left(R_{A},t\right)\;,$$
where
(39)$$\begin{array}{cc} E_{\mathrm{dip},B}\left(R_{A},t\right) & =\frac{1}{4\pi \varepsilon_{0}}\left\{\frac{3\left[\hat{r}\cdot d_{B}\left(t_{r}\right)\right]\hat{r}-d_{B}\left(t_{r}\right)}{r^{3}}+\frac{3\left[\hat{r}\cdot {\dot{d}}_{B}\left(t_{r}\right)\right]\hat{r}-{\dot{d}}_{B}\left(t_{r}\right)}{cr^{2}}\right.\\ \; & +\left.\frac{\left[\hat{r}\cdot {\ddot{d}}_{B}\left(t_{r}\right)\right]\hat{r}-{\ddot{d}}_{B}\left(t_{r}\right)}{c^{2}r}\right\}\;, \end{array}$$
with $r=R_{A}-R_{B}$ and $t_{r}=t-r/c$ being the retarded time. This expression corresponds to the electric field generated by dipole *B* at the position of molecule *A*. This result is readily extended to any number of molecules by exchanging $H_{B}⟶H_{B}+H_{C}+\cdots$ in Equation (35), thus obtaining
(40)$$H_{F}^{\mathrm{eff}}=-d_{A}\left(t\right)\cdot \left[E\left(R_{A},t\right)+\sum_{\zeta =B,C,...}E_{\mathrm{dip},\zeta }\left(R_{A},t\right)\right]\;.$$
While Equation (37) is exact and constitutes an equivalent Hamiltonian, Equation (40) is effective and valid only up to linear order in $d_{B}$, $d_{C}$, etc. It is worth mentioning that Equation (40) can be generalized to other situations. For instance, if atom *A* is in the presence of a magnetically polarizable atom [30,31], we have to add the electric field produced by the magnetic dipole of atom *B* in Equation (38). In the next section, we demonstrate the convenience of the new effective Hamiltonian $H_{F}^{\mathrm{eff}}$ by obtaining the RET rate in a first-order calculation.

## 5. Application to the Resonance Energy Transfer

In a resonance energy transfer (RET) process, an excited molecule decays through nonradiative channels, transferring its energy to a molecule in the ground state [32,33,34,35,36,37,38,39,40,41]. This phenomenon is of notable importance to many areas of science due to its broad range of applications across fields such as chemistry [42], medicine [43], and biology [44]. Throughout this section, we discuss the probability that an excited molecule *A* decays, exciting an identical molecule *B* that was initially in its ground state, placed at a distance *r* from *A*, with both in vacuum.

Up to the second order in perturbation theory, the probability amplitude of interest can be calculated as [15]
(41)$$M_{fi}\approx \langle \psi_{f}|H_{\mathrm{int}}|\psi_{i}\rangle +\lim_{\eta \to 0_{+}}\sum_{r}\frac{\langle \psi_{f}|H_{\mathrm{int}}|\psi_{r}\rangle \langle \psi_{r}|H_{\mathrm{int}}|\psi_{i}\rangle }{E_{i}-E_{r}+i\eta }\;.$$
In this expression, $|\psi_{i}\rangle =|e_{A},g_{B},0_{k\sigma }\rangle$ (with energy $E_{i}$) and $|\psi_{f}\rangle =|g_{A},e_{B},0_{k\sigma }\rangle$ describe, respectively, the system’s initial and final states, ${\hat{H}}_{\mathrm{Int}}$ is the interaction Hamiltonian, and $|\psi_{r}\rangle$ are the intermediate states with energy $E_{r}$. In the standard approach, the interaction Hamiltonian is taken as the dipolar Hamiltonian given by Equation (34): $H_{\mathrm{int}}=H^{\left(2\right)}$. With this choice, however, the first term in Equation (41) vanishes, and the RET rate is obtained from second-order perturbation theory. Here, we offer an alternative and simpler approach by letting atom *B* dress the field and taking $H_{\mathrm{int}}=H_{F}^{\mathrm{eff}}$, as in Equation (40). In this case, it suffices to calculate the first-order matrix element
(42)$$\begin{array}{c} M_{fi}=-\langle g_{A}|d_{A}\left(t\right)|e_{A}\rangle \cdot \langle e_{B}|E_{\mathrm{dip},B}\left(R_{A},t\right)|g_{B}\rangle \;. \end{array}$$
Following Equation (Equation 21), the first matrix element on the right-hand side of the previous equation becomes (more precisely, we should consider the evolution beginning at time $t_{0}$; however, as explained in Section 3, this would only contribute as an irrelevant global phase)
(43)$$\begin{array}{c} \langle g_{A}|d_{A}\left(t\right)|e_{A}\rangle =e^{-i\omega_{eg}t}d_{A}^{ge}\;, \end{array}$$
and, by using Equation (39), the terms contained in the second matrix element give the contributions
(44)$$\begin{array}{cc} \langle e_{B}|d_{B}\left(t_{r}\right)|g_{B}\rangle & =e^{i\omega_{eg}t}e^{-ikr}d_{B}^{eg}\;,\\ \langle e_{B}|\partial_{t}d_{B}\left(t_{r}\right)|g_{B}\rangle & =\frac{\partial }{\partial t}\langle e_{B}|d_{B}\left(t_{r}\right)|g_{B}\rangle =i\omega_{eg}e^{i\omega_{eg}t}e^{-ikr}d_{B}^{eg}\;, \end{array}$$
(45)$$\begin{array}{cc} \langle e_{B}|\partial_{t}^{2}d_{B}\left(t_{r}\right)|g_{B}\rangle & =\frac{\partial^{2}}{\partial t^{2}}\langle e_{B}|d_{B}\left(t_{r}\right)|g_{B}\rangle =-\omega_{eg}^{2}e^{i\omega_{eg}t}e^{-ikr}d_{B}^{eg}\;, \end{array}$$
where $k=\omega_{eg}/c$. Replacing these results in Equation (42), we arrive at
(46)$$M_{fi}=\frac{d_{i,A}^{ge}d_{j,B}^{eg}e^{-ikr}}{4\pi \epsilon_{0}r^{3}}\left[\left(\delta_{ij}-3{\hat{r}}_{i}{\hat{r}}_{j}\right)\left(1+ikr\right)-\left(\delta_{ij}-{\hat{r}}_{i}{\hat{r}}_{j}\right)k^{2}r^{2}\right]\;.$$
By applying Fermi’s golden rule,
(47)$$\Gamma^{\mathrm{RET}}=\frac{2\pi }{\hbar^{2}}\rho \left(\omega_{f}\right){\left|M_{fi}\right|}^{2}\;,$$
where $\rho (\omega_{f})$ is the density of final states with energy $E_{f}=\hbar \omega_{f}$, and we directly recover the well-known result for the RET rate [45,46].

## 6. The Dressed Molecules Dress the Field—And the Reverse

The first-order perturbation in the Hamiltonian (40) vanishes whenever the electric dipole operator of one of the molecules cannot connect the involved molecular states. An example is the force between molecules in their ground state to be analyzed in the next section. Here, instead, we focus on a general discussion without specifying the molecular internal state. The physical mechanism that limits the dipole–dipole correlation depends strongly on the distance *R* separating the molecules. Indeed, two characteristic time scales are key to understanding the two different regimes: the time it takes light to travel between the molecules, $t_{\gamma }=r/c$, and the characteristic time for dipole fluctuations, $t_{d}=1/\omega_{0}$, where $\omega_{0}$ is a typical transition frequency for the molecules. In the asymptotic long-distance regime, $t_{\gamma }\gg t_{d}$, it is the electrodynamical retardation that limits the dipole–dipole correlation, and we may neglect dispersion in the atomic response. In the opposite short-distance regime, electrodynamical retardation is negligible, and it is now the delay in the molecular response that limits the dipole–dipole correlation. Now, the molecular dispersion is crucial, but we can take the electrostatic limit for the electric field produced by each electric dipole. To go deeper into the physical particularities of these two complementary regimes, we shall develop a different effective Hamiltonian appropriate to each case.

### 6.1. The Dressed Molecules Dress the Field: Retarded Long-Distance Regime

In the long-distance regime, we may neglect dispersion in the molecules, which is tantamount to considering an instantaneous molecular response. This means that the time-scale variation for the electric field is much slower than the molecular response, enabling us to approximate
(48)$$\frac{i}{\hbar }\int_{-\infty }^{t}dt^{\prime }\left[d_{j}\left(t\right),d_{l}\left(t^{\prime }\right)\right]E_{l}\left(R,t^{\prime }\right)\approx \Pi_{jl}E_{l}\left(R,t\right)\;,$$
when evaluating the effective Hamiltonian $H_{M}^{\mathrm{eff}}\left(t\right)$ given by Equation (Equation 11). We have defined the molecular operator $\Pi_{jl}$ as
(49)$$\Pi_{jl}=\frac{i}{\hbar }\int_{-\infty }^{\infty }dt^{\prime }\theta \left(t-t^{\prime }\right)\left[d_{j}\left(t\right),d_{l}\left(t^{\prime }\right)\right]\;.$$
From Equation (Equation 13), we see that the expectation value $\langle g|\Pi_{jl}|g\rangle$ is the static polarizability of the molecule in its ground state, given by setting ω=0 in Equation (Equation 18). The tensor operator $\Pi_{jl}$ generalizes this concept by enabling us to capture the static response even for processes involving changes in the molecular internal state. Substituting (48) into (Equation 11) leads to
(50)$$H_{M}^{\mathrm{eff},s}\left(t\right)=-\frac{1}{2}\Pi_{jl}E_{j}\left(R,t\right)E_{l}\left(R,t\right)\;,$$
which corresponds to the static response limit of $H_{M}^{\mathrm{eff}}\left(t\right)$.

In the case of two molecules, dipole–dipole correlations arise in second-order perturbation theory in the Hamiltonian
(51)$$H_{M}^{\mathrm{eff},s_{\left(2\right)}}\left(t\right)=H_{M,A}^{\mathrm{eff},s_{\left(2\right)}}\left(t\right)+H_{M,B}^{\mathrm{eff},s_{\left(2\right)}}\left(t\right)=-\frac{1}{2}\Pi_{A,jl}E_{j}\left(R_{A},t\right)E_{l}\left(R_{A},t\right)-\frac{1}{2}\Pi_{B,jl}E_{j}\left(R_{B},t\right)E_{l}\left(R_{B},t\right)\;,$$
where $\Pi_{\zeta ,jl}$ is the operator (49) for molecule ζ=A,B.

An equivalent Hamiltonian where the molecules couple directly with each other will be able to capture the dipole–dipole correlation in first-order perturbation theory. This can be done by employing the unitary transformation
(52)$$U_{\mathrm{MF}}=\tilde{T}e^{\frac{i}{\hbar }\int_{-\infty }^{t}\;\;\;\;dt^{\prime }H_{M,B}^{\mathrm{eff},s}\left(t^{\prime }\right)}\;,$$
which mimics the one employed in Section 4, with the difference that it is the dressed molecule (through operator $\Pi_{jl}$), instead of the naked molecule, that dresses the field. Following the same steps that led us from Equation (34) for molecule *B* into Equation (37), we get
(53)$$H_{\mathrm{MF}}^{\mathrm{eff},s}\left(t\right)=U_{\mathrm{MF}}H_{M,A}^{\mathrm{eff},s}\left(t\right)U_{\mathrm{MF}}^{-1}\;.$$
$H_{\mathrm{MF}}^{\mathrm{eff},s}$ is very suitable for handling effects related to the interaction between atoms *A* and *B* because the expansion of $U_{\mathrm{MF}}$ in Equation (53) contains terms combining the product $\Pi_{A,jl}\Pi_{B,mn}$. Such terms also appear through a fourth-order perturbation theory in Hamiltonian (34) but already appear in first order here. To obtain $H_{\mathrm{MF}}^{\mathrm{eff},s}$, it is enough to implement the transformation rule for the electric field at $R_{A}$, since $\Pi_{A,jl}$ commutes with $\Pi_{B,mn}$. Substituting Equation (50) into (52), we obtain the following up to linear order in $\Pi_{B}$ (see Appendix A):(54)$$U_{\mathrm{MF}}E\left(R_{A},t\right)U_{\mathrm{MF}}^{-1}\approx E\left(R_{A},t\right)+E_{\mathrm{dip},B}^{\mathrm{ind}}\left(R_{A},t\right)\;,$$
where $E_{\mathrm{dip},B}^{\mathrm{ind}}\left(R_{A},t\right)$ is given by Equation (39) with the substitution $d_{B,j}\left(t_{r}\right)⟶\Pi_{B,jk}E_{k}\left(R_{B},t_{r}\right)$. This result is mathematically similar to Equation (38) but with a remarkable physical difference. Here, the field is dressed not by a naked molecule but by a dressed one. This means that the source of the electric dipole field is not the molecular dipole operator but rather a vacuum-induced dipole—as indicated by the superscript “ind” in Equation (54). From Equations (53) and (54), we arrive at
(55)$$H_{\mathrm{MF}}^{\mathrm{eff},s}\left(t\right)=-\frac{1}{2}\Pi_{A,jk}\left(E_{k}\left(R_{A},t\right)E_{\mathrm{dip},B,j}^{\mathrm{ind}}\left(R_{A},t\right)+E_{\mathrm{dip},B,k}^{\mathrm{ind}}\left(R_{A},t\right)E_{j}\left(R_{A},t\right)\right)\;,$$
where we have kept only the terms capturing the dipole–dipole correlation and neglected the higher-order term $\Pi_{A,jk}E_{\mathrm{dip},B,k}^{\mathrm{ind}}\left(R_{A},t\right)E_{\mathrm{dip},B,j}^{\mathrm{ind}}\left(R_{A},t\right)$. This expression is manifestly symmetric upon the exchange A⟷B, as can be verified by substituting Equation (39) into (55).

As an application of the effective Hamiltonian (55), we may consider that both molecules are at their ground states throughout the entire process. In this case, we may take the expectation value of $H_{\mathrm{MF}}^{\mathrm{eff},s}\left(t\right)$ in the ground states of the molecules, which is equivalent to substituting the operators $\Pi_{jl}$ with the static polarizability tensor of the corresponding molecule in Equation (55). In the particular case of isotropic molecules, this reproduces the asymptotic long-distance limit of the effective Hamiltonian originally employed by P.W. Milonni, which readily yields the known Casimir–Polder result in first order [18]. For comparison, when taking the average of $H_{M}^{\mathrm{eff},s_{\left(2\right)}}$ (see Equation (51)) over the molecular ground state, the resulting effective Hamiltonian [47] yields the Casimir–Polder energy only to the second order of perturbation theory.

In the opposite short-distance limit, molecular dispersion is essential, and we cannot make the approximation given by Equation (48). In this case, it is more convenient to work with a different effective Hamiltonian, which we shall present in the next subsection.

### 6.2. The Dressed Field Dresses the Molecules: Non-Retarded Regime

We now turn to the opposite regime, in which the intermolecular distance is so small that we may neglect the electromagnetic retardation in comparison to the molecular response time. Unlike the other examples in this paper, this case does not require quantization of the electromagnetic field. On the other hand, the molecular dispersion is crucial in this regime. The dipole–dipole correlation is usually obtained from a second-order perturbation theory in the dipole–dipole Hamiltonian [48]
(56)$$H_{dd}=\frac{d_{A,j}d_{B,k}\left(\delta_{jk}-3{\hat{r}}_{j}{\hat{r}}_{k}\right)}{4\pi \varepsilon_{0}r^{3}}\;,$$
where $r=r\hat{r}$ is the position of molecule *B* with respect to molecule *A*. Notice that this Hamiltonian is a particular case of Equation (40) without the vacuum electric field operator and with the dipole electric field taken in the electrostatic approximation. In the non-retarded regime, the field and the molecular operators switch roles with respect to what happens in the retarded asymptotic regime. While, in the latter, we could begin with Hamiltonian (48), which approximates the molecular response with its dressed static response, Equation (56) is precisely the opposite: now it is the field that is dressed by its static response. Indeed, the electrostatic dipole field leading to (56) corresponds to the zero-frequency limit of the Green function of the wave equation, which plays the role of the field susceptibility, as discussed in Appendix B.

Mirroring the procedure of the previous subsection, we let the already-dressed field dress the molecules, thus leading to a new effective Hamiltonian. In the long-distance regime, we employed a unitary transformation extending the formalism of Section 4. Here, on the other hand, we want to extend the formalism developed in Section 2. As a first step, we write Hamiltonian (56) in the interaction picture and then take as the unitary transformation the operator
(57)$$U_{\mathrm{FM}}=\exp\left(\frac{i}{\hbar }\int_{-\infty }^{t}\;\;\;\;dt^{\prime }H_{dd}\left(t^{\prime }\right)\right)\;,$$
which should be compared to Equation (Equation 5). Following steps analogous to the ones leading to Equation (Equation 10) yields
(58)$$H_{\mathrm{FM}}^{\mathrm{eff}}\left(t\right)=-\frac{i}{2\hbar }\int_{-\infty }^{t}\;\;\;\;dt^{\prime }\left[H_{dd}\left(t\right),H_{dd}\left(t^{\prime }\right)\right]\;.$$
Since operators involving different molecules commute, we have
(59)$$\begin{array}{cc} \left[d_{A,j}\left(t\right)d_{B,k}\left(t\right),d_{A,m}\left(t^{\prime }\right)d_{B,n}\left(t^{\prime }\right)\right] & =\frac{\left[d_{A,j}\left(t\right),d_{A,m}\left(t^{\prime }\right)\right]\left\{d_{B,k}\left(t\right),d_{B,n}\left(t^{\prime }\right)\right\}}{2}\\ \; & +\frac{\left\{d_{A,j}\left(t\right),d_{A,m}\left(t^{\prime }\right)\right\}\left[d_{B,k}\left(t\right),d_{B,n}\left(t^{\prime }\right)\right]}{2}\;. \end{array}$$
By substituting Equation (56) and the previous identity into Equation (58), we obtain the effective Hamiltonian
(60)$$\begin{array}{cc} H_{\mathrm{FM}}^{\mathrm{eff}} & =-\frac{i\left(\delta_{jk}-3{\hat{r}}_{j}{\hat{r}}_{k}\right)\left(\delta_{mn}-3{\hat{r}}_{m}{\hat{r}}_{n}\right)}{64\hbar \pi^{2}\varepsilon_{0}^{2}r^{6}}\int_{-\infty }^{t}\;\;\;\;dt^{\prime }(\left[d_{A,j}\left(t\right),d_{A,m}\left(t^{\prime }\right)\right]\left\{d_{B,k}\left(t\right),d_{B,n}\left(t^{\prime }\right)\right\}\\ \; & +\left\{d_{A,j}\left(t\right),d_{A,m}\left(t^{\prime }\right)\right\}\left[d_{B,k}\left(t\right),d_{B,n}\left(t^{\prime }\right)\right])\;. \end{array}$$

The new effective Hamiltonian (60) has two terms that capture the physics involved in the dipole–dipole correlation. The product $\left[d_{A,j}\left(t\right),d_{A,m}\left(t^{\prime }\right)\right]\left\{d_{B,k}\left(t\right),d_{B,n}\left(t^{\prime }\right)\right\}$ measures how the dipole fluctuations of molecule *B* induces a dipole in molecule *A*, while the other term is its reciprocal. This decomposition is possible because, differently from the standard approach based on second-order perturbation theory with the time-independent dipole–dipole Hamiltonian (56) where the two molecules are considered as an isolated system, here, we take the complementary approach of considering each molecule separately as an open quantum system. This perspective offers two main novelties: (i) $H_{\mathrm{FM}}^{\mathrm{eff}}$ brings to light the dynamical character of the dispersion interaction by making an explicit connection with dipole fluctuations, and (ii) $H_{\mathrm{FM}}^{\mathrm{eff}}$ enables us to assess the contribution from the fluctuations of each molecule separately. In the next section, we analyze an example that illustrates these advantages.

## 7. Application to the London Interaction Energy

In this section, we consider the dispersion interaction between two ground-state nonpolar molecules *A* and *B* in vacuum, which interact due to correlations between their fluctuating electric dipoles. As discussed in the previous section, the physical mechanism limiting the dipole–dipole correlation strongly depends on comparing the distance separating the molecules and their internal transition wavelengths. For ground-state molecules, the resulting intermolecular interaction energy exhibits a different power-law dependence with distance in each of the two regimes discussed in Section 6.

As originally demonstrated by London [49], the non-retarded interaction energy can be obtained without quantizing the electromagnetic field and scales with $1/r^{6}$. The asymptotic long-distance limit was first obtained in the seminal paper by Casimir and Polder [50], where they showed that retardation imposes the necessity of quantizing the electromagnetic field and demonstrated that the interaction energy scales asymptotically with $1/r^{7}$.

Both regimes have still been at the center of intense investigation in recent years. Casimir–Polder forces have been studied considering excited [51,52,53,54,55,56] and chiral [57,58,59,60] particles. The influence of neighboring surfaces with ever-increasing complexity [61,62,63,64,65,66,67,68,69,70,71,72,73] and with dynamical [74,75,76] and thermal effects [77,78,79,80,81,82,83] has also been considered.

The force in the non-retarded regime—sometimes referred to as London or van der Waals force—plays a pivotal role in chemistry [84] and condensed matter physics, where short-range interactions prevail. In van der Waals heterostructures, two-dimensional materials are stacked and held together by London dispersion forces, generating materials with fascinating physical properties that are useful for designing new electronic devices [85,86,87]. Density functional theory provides a powerful framework capable of obtaining increasingly precise descriptions of molecular polarizabilities and London dispersion forces [88,89,90]. Modifications of the force due to an intervening electrolyte medium [91,92,93], with the atomic motion in connection with quantum friction [94,95,96,97,98,99,100,101,102,103,104,105] or with non-local interferometric phases [106,107,108], the atomic internal state [109], and coming from boundary conditions imposed by nearby structures [110,111,112,113,114], have disclosed important features of the London–van der Waals interactions.

In the previous section, we discussed how the Casimir–Polder result for the asymptotic long-distance limit can be derived from the effective Hamiltonian $H_{\mathrm{MF}}^{\mathrm{eff}}$ (55). In this section, we obtain the London result for the short-distance non-retarded limit from the new effective Hamiltonian $H_{\mathrm{FM}}^{\mathrm{eff}}$ (60), which provides new physical insights into the dipole–dipole correlation present in the non-retarded regime. By taking the average of Hamiltonian (60) in the ground state of the molecules and employing the result (Equation 13) for the molecular polarizability tensors $\alpha_{jl}^{A},\alpha_{jl}^{B}$, we obtain
(61)$$E_{\mathrm{London}}=-\frac{\hbar \left(\delta_{jk}-3{\hat{r}}_{j}{\hat{r}}_{k}\right)\left(\delta_{mn}-3{\hat{r}}_{m}{\hat{r}}_{n}\right)}{64\pi^{2}\varepsilon_{0}^{2}r^{6}}\int_{-\infty }^{\infty }\;\;\;d\tau \left(\alpha_{jm}^{A}\left(\tau \right)\eta_{kn}^{B}\left(\tau \right)+\eta_{jm}^{A}\left(\tau \right)\alpha_{kn}^{B}\left(\tau \right)\right),$$
where we defined the symmetrical dipole correlation function
(62)$$\eta_{jm}^{\zeta }\left(\tau \right):=\frac{1}{\hbar }\langle g|\left\{d_{j}^{\zeta }\left(t^{\prime }+\tau \right),d_{m}^{\zeta }\left(t^{\prime }\right)\right\}|g\rangle \;,$$
with ζ=A,B. To work in the Fourier space, we can apply Parseval’s theorem, so Equation (61) becomes
(63)$$E_{\mathrm{London}}=-\frac{\hbar \left(\delta_{jk}-3{\hat{r}}_{j}{\hat{r}}_{k}\right)\left(\delta_{mn}-3{\hat{r}}_{m}{\hat{r}}_{n}\right)}{128\pi^{3}\varepsilon_{0}^{2}r^{6}}\int_{-\infty }^{\infty }\;\;\;d\omega \left(\alpha_{jm}^{A}\left(\omega \right)\eta_{kn}^{B}\left(\omega \right)+\eta_{jm}^{A}\left(\omega \right)\alpha_{kn}^{B}\left(\omega \right)\right),$$
where we used that $\eta^{A,B}\left(\omega \right)$ are real functions, as they are Fourier transforms of real even functions. This result is the analog for two molecules of the decomposition obtained for an atom coupled to the vacuum electric field [115,116,117]. In the latter, the field susceptibility captures the field radiation reaction. More recently, an analogous decomposition was also obtained for atoms interacting with a scalar quantum field [118]. A decomposition similar to (63) was employed to derive a nonlocal phase for a moving atom interacting with a planar surface [119] and a Sagnac-like atomic phase induced by a rotating nanosphere [120].

In the isotropic case, the polarizability tensors simplify to $\alpha_{rs}^{A(B)}\left(\tau \right)=\delta_{rs}\alpha^{A(B)}\left(\tau \right)$, and the symmetric correlation functions simplify to $\eta_{rs}^{A(B)}\left(\tau \right)=\delta_{rs}\eta^{A(B)}\left(\tau \right)$. Then, Equation (63) leads to
(64)$$E_{\mathrm{London}}=-\frac{3\hbar }{64\pi^{3}\varepsilon_{0}^{2}r^{6}}\int_{-\infty }^{\infty }d\omega \left(\alpha^{A}\left(\omega \right)\eta^{B}\left(\omega \right)+\eta^{A}\left(\omega \right)\alpha^{B}\left(\omega \right)\right)\;.$$
In Appendix B, we employ the analytical properties of the correlation functions to demonstrate that our results are equivalent to the standard way of expressing the London interaction energy for any molecular model of the polarizabilities. Here, we show the convenience of Equation (64) by considering the simple case of two-level atoms, for which (ζ=A,B) [18]
(65)$$\begin{array}{cc} \alpha^{\zeta }\left(\omega \right) & =\frac{\alpha_{0}^{\zeta }\omega_{0\zeta }^{2}}{\omega_{0\zeta }^{2}-\omega^{2}}\;, \end{array}$$
(66)$$\begin{array}{cc} \eta^{\zeta }\left(\omega \right) & =\pi \alpha_{0}^{\zeta }\omega_{0\zeta }\left[\delta \left(\omega -\omega_{0\zeta }\right)+\delta \left(\omega +\omega_{0\zeta }\right)\right]\;. \end{array}$$
Let us analyze each contribution to the London interaction energy in Equation (64) separately. We define
(67)$$\begin{array}{cc} E_{\mathrm{London}}^{A\to B} & =-\frac{3\hbar }{64\pi^{3}\varepsilon_{0}^{2}r^{6}}\int_{-\infty }^{\infty }d\omega \;\eta^{A}\left(\omega \right)\alpha^{B}\left(\omega \right)\;, \end{array}$$
as the contribution arising from the dipole induced at atom *B* by the dipole fluctuations of atom *A*. From Equations (65) and (66), we obtain
(68)$$\begin{array}{cc} E_{\mathrm{London}}^{A\to B} & =-\frac{3\hbar \alpha_{0}^{A}\alpha_{0}^{B}\omega_{0A}\omega_{0B}}{32\pi^{2}\varepsilon_{0}^{2}r^{6}}\frac{\omega_{0B}}{\omega_{0B}^{2}-\omega_{0A}^{2}}\;. \end{array}$$
The interaction energy is $E_{\mathrm{London}}=E_{\mathrm{London}}^{B\to A}+E_{\mathrm{London}}^{A\to B}$, with $E_{\mathrm{London}}^{B\to A}$ being obtained by interchanging the roles of *A* and *B* in Equation (68).

Let us consider $\omega_{0A}>\omega_{0B}$. In that case, from Equation (68), we see that $E_{\mathrm{London}}^{A\to B}>0$, indicating that the dipole induced at a slower atom by a faster one generates a repulsive contribution to the dispersion force. This is due to the fact that the polarizability given by Equation (65) becomes negative for frequencies higher than the atomic transition frequency. Indeed, the induced dipole at the slower atom *B* cannot follow the fast oscillation of the fluctuating dipole of atom *A*. The induced dipole at *B* lags behind the field of atom *A* and points opposite to its direction at a given time. As a consequence, the induced dipole at *B* repels the fluctuating dipole at *A*. However, the opposite holds for the complementary term $E_{\mathrm{London}}^{B\to A}$: the dipole induced in the faster atom *A* can follow the dipole fluctuations of atom *B* in phase, leading to an attractive contribution. Attraction overcomes repulsion by a factor $\omega_{0A}/\omega_{0B}$, since the slower atom couples less effectively to the field than the faster one. If $\omega_{0A}=\omega_{0B}$, each contribution diverges due to a resonant response. This divergence would be avoided if dissipation were taken into account. Nevertheless, it is remarkable that the divergence cancels once we sum $E_{\mathrm{London}}^{B\to A}$ and $E_{\mathrm{London}}^{A\to B}$, leaving us with the well-behaved total interaction energy
(69)$$E_{\mathrm{London}}=-\frac{3\hbar \alpha_{0}^{A}\alpha_{0}^{B}}{32\pi^{2}\varepsilon_{0}^{2}r^{6}}\frac{\omega_{0A}\omega_{0B}}{\omega_{0A}+\omega_{0B}}\;,$$
which agrees with the result [2] calculated from second-order perturbation theory based on the dipole–dipole Hamiltonian (56).

Notice that varying $\omega_{0B}$ while keeping the other parameters fixed shows that the attraction is maximal when $\omega_{0B}\to \infty$. The previous decomposition clearly illustrates the physical mechanism involved. From Equation (68), we see that in this limit, the repulsive contribution $E_{\mathrm{London}}^{B\to A}$ vanishes, indicating that atom *A* is effectively transparent, decoupling from the rapid oscillating field produced by *B*. The attractive term in Equation (68), on the other hand, takes its maximal absolute value in this limit, since the response of atom *B* is so fast that it perfectly mirrors the fluctuations of the other atom. In this sense, we may conclude that atom *B* in the limit $\omega_{0B}\to \infty$ is the atomic analog of a perfect conductor.

As was true with the other effective Hamiltonians discussed in this paper, we see that the convenience of employing $H_{\mathrm{FM}}^{\mathrm{eff}}$ is twofold: (i) it lowers the perturbation order required to obtain the London dispersion energy from second to first order, and (ii) it offers physical insights into the mechanisms involved in the phenomenon. The results in this section can be readily extended for multilevel atoms. To this end, it suffices to substitute Equations (65) and (66) with a summation over all internal transition frequencies.

## 8. Final Remarks and Conclusions

All phenomena in molecular quantum electrodynamics can be obtained from the multipolar Hamiltonian. In this paper, we restricted our attention to phenomena that can be treated perturbatively (which includes the vast majority of cases in this field). In most situations, the dominant effect is obtained from a high-order perturbation theory, requiring intermediate states to connect the initial and the final states. A clear example is the interatomic interaction. While in classical electrodynamics, we may always take the field at each charged particle as the superposition of the field generated by all other particles, in standard quantum electrodynamics, each particle couples only to the free electromagnetic field. Consequently, we must go up to the fourth order to obtain the dominant contribution when considering molecules without permanent electric dipole moments. An alternative is to build effective Hamiltonians. They are customized for each specific application and lose meaning and validity after some point in the perturbative expansion. Not only do they greatly simplify the technical difficulties involved in calculations using the multipolar Hamiltonian, but, equally importantly, the effective Hamiltonians cast the phenomena in a new light, offering insightful physical interpretations.

Several effective Hamiltonians have successfully been employed by many authors in the last decades. In this paper, we have developed a systematic approach to constructing effective Hamiltonians, which allowed us to derive a number of new ones, choosing as a unitary transformation the Hermitian conjugate of the evolution operator for part of the system. This transfers part of the time evolution from the vector state to the operators, dressing them and providing a Hamiltonian that requires a lower order in perturbation theory to account for the process of interest. This method can always be used when the first-order perturbation theory vanishes. Our approach yields time-dependent Hamiltonians that enable us to follow the energy exchange between matter and the field, with each subsystem constituting an open quantum system. We emphasize here that our system of interest is the entire molecule–field system, and it is not interesting to trace over any of the subsystems, as is common in an open quantum system approach [121,122,123].

As a first application, we have derived the Hamiltonian $H_{M}^{\mathrm{eff}}$, where the field dresses the molecule and the dipole operators are replaced by their commutator at different times. If we project the commutator in an internal molecular state, it yields the dynamical polarizability of the molecule for the corresponding state. Its nondiagonal elements, on the other hand, allow the dressing to leave the molecule in a final state that is different from the initial one. We have demonstrated that this new time-dependent Hamiltonian provides a simpler treatment of the two-photon spontaneous emission, as the dominant contribution is obtained in first-order perturbation theory. In addition, our formalism introduces the concept of an induced dipole transition, which generalizes the notion of an induced dipole for a given internal state.

Then, we discussed applications involving two molecules *A* and *B*. We constructed the new effective Hamiltonian $H_{F}^{\mathrm{eff}}$ through a unitary transformation that transfers all of the effects related to molecule *B* to the electric field it generates. In this way, molecule *A* feels an effectively dressed electric field given by the superposition of the free vacuum electric field, and the one generated by the dipole operator of molecule *B*. $H_{F}^{\mathrm{eff}}$ allows for the description of the resonance energy transfer rate in first-order perturbation theory.

Lastly, we derived two additional Hamiltonians that merge aspects of the previous two, where each one is appropriate for a different intermolecular distance regime. In the asymptotic long-distance regime, molecular dispersion is negligible, enabling us to derive an effective Hamiltonian $H_{\mathrm{MF}}^{\mathrm{eff}}$ which is formally similar to $H_{M}^{\mathrm{eff}}$. In this new case, however, the field acting on molecule *A* is given by the superposition of the free electric field and the one produced by the vacuum-induced dipole generated on molecule *B*. When we average $H_{\mathrm{MF}}^{\mathrm{eff}}$ over the molecular ground state, we re-obtain the asymptotic limit of the Hamiltonian employed by P. Milonni [18].

Finally, for the short-distance non-retarded limit, we derived our fourth and last effective Hamiltonian $H_{\mathrm{FM}}^{\mathrm{eff}}$ based on the fact that, in this limit, we do not need to quantize the electric field. This effective Hamiltonian enables us to clearly identify the different physical mechanisms involved in the correlations responsible for the interaction, separating one term where the dipole fluctuations of molecular *A* induce a dipole on molecule *B* and another term where the roles are exchanged.

As a final application, we employed $H_{\mathrm{FM}}^{\mathrm{eff}}$ to obtain the London dispersion interaction energy in first-order perturbation theory. We showed that, for two-level atoms, the dipole fluctuations of the atom with the higher transition frequency give rise to a repulsive term, since its fast fluctuations cannot be followed by the slower atom. Nonetheless, the force between two isotropic atoms is always attractive, since the fluctuations of the slower atom are strongly correlated and easily followed by the faster atom, overcoming the repulsive contribution. The possibility of quantitatively and separately analyzing the contributions arising from each mechanism correlating fluctuating systems is an advantage of $H_{\mathrm{FM}}^{\mathrm{eff}}$.

The Hamiltonians presented in this paper can be employed in a great variety of situations. For instance, one may treat the effects of boundaries in the two-photon spontaneous emission or resonance energy transfer by simply introducing the appropriate field modes. As in the examples discussed in this paper, these effective Hamiltonians allow for a direct first-order calculation within perturbation theory. More notably, the methodology introduced here can be applied to generate other effective Hamiltonians that may optimize calculations and provide physical intuition.

## Data Availability

The original contributions presented in the study are included in the article, further inquiries can be directed to the corresponding author/s.

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
