# Peer review of "Time-Dependent Effective Hamiltonians for Light–Matter Interactions"

_entropy, 2024, doi:10.3390/e26060527_

Round 1
Reviewer 1 Report
Comments and Suggestions for Authors
The manuscript of Aroaldo Santos is a theoretical work in the field of quantum optics and electrodynamics.
The authors propose a systematic approach for building effective Hamiltonians in molecular quantum electrodynamics.
After quite extensive introduction in Sec.1, the authors proceed in the following order: derivation of the effective Hamiltonian for a neutral molecule dressed by the quantized electromagnetic field and obtain the so-called dynamical polarizability Hamiltonian (Eq. 16). In Sec. 3, they study the two-photon spontaneous emission using this method and obtain the rate of the process in Eq. (29). In Sec. 4, the authors consider the reverse effect, namely, dressing of the electromagnetic field by molecules and obtain an effective Hamiltonian (Eq. 39). In Sec. 5, this Hamiltonian is employed to study the so-called resonance energy transfer (RET) process between two quantum emitters, and the respective transition matrix element is derived in Eq. (45). Sections 6 and 7 contain further application of the method. Thus, in Sec. 6, the authors consider the retarded and non-retarded regimes of inter-molecule interactions, and obtain Hamiltonians, Eqs. (53) and (58). From the retarded regime the Casimir-Polder energy can be derived. In the final Sec. 7, the dispersion interaction between two ground-state nonpolar molecules is considered and the London energy is derived.
The manuscript is written in clear and logical technical English, all derivations are well explained, and different technical terms are properly introduced. The main merit of the manuscript is methodological. The authors paid a lot of attention to properly introduce and motivate their method. I see only one weak point of the manuscript, namely, that no new applications are presented. However, this can be tolerated---one needs to first establish a formalism, and only after that apply it to new problems. Therefore, I recommend publication of the manuscript in the present form.
However, I would also like to encourage the authors to open new avenues for their study.
Reviewer 2 Report
Comments and Suggestions for Authors
Review: Time-dependent effective Hamiltonians for light-matter interactions
The paper is generally well written. Unfortunately, the tight timeframe set by the journal does not allow for a proper review in which the equations are re-derived and a detailed evaluation of a 23-page article is conducted. Nevertheless, we would like to provide (at least some) feedback, as there are several points we think the authors should address before final publication.
General questions:
I) The gauge in which the electromagnetic potentials are written has never been defined. This should be done rigorously.
II) The article would benefit from a more detailed explanation of when the authors' approach is applicable. This is only briefly done on page 5. In particular, what is the advantage over regular perturbation theory, where e.g. all odd terms vanish?
III) The authors write that the great advantage of their approach is that it describes physics as an open quantum system. This begs the question why they did not derive a master equation with the density matrix or the density operator in the first place. The authors should comment on this.
IV) The summary is very strongly worded. The style of writing makes clear that the authors want to promote their results. However, they completely overlook all non-perturbative physics. I therefore strongly encourage the authors to rewrite parts of section 8. Especially at the beginning. In this context, they also write "particles couple only to the free electromagnetic field", which is generally not true.
Additional remarks:
Equation (1): For the sake of completeness, it would be good to see the free Hamiltonian H0.
Equations (7-8): I think the second term is wrong. It should be a Dyson or Neumann series, and thus a double integral. Here one might get the impression that a single integral is evaluated and then the result is squared.
Equation (11): The point that the operator averaging is done only within the subspace of the molecule should be more clearly pointed out. Otherwise it could be confusing why the Hamiltonian on the left is still an operator.
Section 3, first sentence: I think this sentence is not quite correct. Positronium is the simplest molecule imaginable, and it already has two different states. One of them decays mainly into three photons.
Equation (18): It should be made clear why the spatial dependence in the field is neglected.
Equation (24): Something went wrong with the formatting of this equation.
Equation (26): Something went wrong with the formatting of this equation. I think it should read "dt' H(t')".
Equation (38): Normally the capital "R" is reserved for the centre of mass coordinate, not the relative coordinate.
Page 10 - Line 308: Last sentence of section 5. It would be beneficial to have the result for the RET rate in this paper.
Page 11 - Line 381: The references appear to be out of order.
Page 15 - Line 500: "...it lowers to first...". Something has gone wrong here as this is not a correct sentence.
